# The HD-ZIP II Transcription Factors HAT3 and ATHB4 Fine-Tune Auxin and Cytokinin Pathways During Flower Development

**DOI:** 10.3390/plants14243723

**Published:** 2025-12-06

**Authors:** Kestrel A. Maio, Sophia Luche, Monica Carabelli, Laila Moubayidin

**Affiliations:** 1John Innes Centre, Norwich Research Park, Colney Ln, Norwich NR4 7UH, UK; kestrel.maio@jic.ac.uk; 2Department of Chemistry, Life Sciences and Environmental Sustainability, University of Parma, Parco Area delle Scienze 11/A, 43124 Parma, Italy; sophia.luche@unipr.it; 3Institute of Molecular Biology and Pathology, National Research Council, P.le A. Moro 5, 00185 Roma, Italy; monica.carabelli@cnr.it

**Keywords:** HD-ZIPs class-II, auxin, cytokinin, hormonal homeostasis, flower development, trichome patterning, phyllotaxy, transcription factors, plant reproduction

## Abstract

Flowers are key reproductive structures for many plant species. They are essential for seed and fruit production, and their development is tightly regulated by hormonal and genetic networks. The homeodomain transcription factors HAT3 and ATHB4 are known regulators of adaxial identity and hormone response. We demonstrate that flowers of the *hat3 athb4* double mutant emerge at wider divergence angles relative to the wild type, a phenotype reflecting modified phyllotaxy and regulated by low auxin conditions. In addition, *hat3 athb4* flowers exhibit aberrant trichome patterning on their sepals associated with enhanced sensitivity to cytokinin (CK). Through RNA-seq analysis of *hat3 athb4* inflorescences, we identify the misregulation of genes involved in auxin biosynthesis (*YUCCAs*), auxin transport (*PID*), and CK metabolism (*CKXs*) and transport (*PUPs*). These findings suggest that HAT3 and ATHB4 fine-tune the auxin/CK balance and coordinate critical pattern events during reproductive development, offering new insight into hormone-mediated regulation of floral patterning.

## 1. Introduction

Flowers are the reproductive organs of angiosperms, playing a central role in the continuation of plant species. They are not only essential for the propagation of many crops but are directly responsible for the production of grains, fruits, and seeds that form the foundation of the human diet. A deeper understanding of floral development and function holds tremendous potential for agricultural innovation. Among the key regulators of flower formation are the phytohormones auxin and cytokinin (CK), which coordinate crucial developmental processes such as organ initiation, meristem activity, cell differentiation, and patterning within floral tissues [1,2,3,4]. Auxin gradients, for instance, help to determine the positioning of floral organs, while CK modulates meristem size, cell proliferation, and differentiation during trichome development [3,4,5,6,7,8]. Deciphering the molecular mechanisms and regulatory networks governed by these hormones offers valuable insights into how flower development can be manipulated. This knowledge may enable the targeted enhancement of reproductive traits, such as flower number, fertility, and fruit yield [9,10]. Such advances are particularly vital as modern agriculture grapples with the dual challenge of feeding a growing population and coping with increasingly unpredictable climatic conditions and shortages of fertilisers. In this context, flowers are not merely reproductive structures—they are a gateway to sustainable crop improvement, and hormone-guided strategies could be key to ensuring food security in the decades to come.

Flower organ identity, positioning, growth, and polarity are governed by a complex network of genetic interactions and hormonal signalling pathways. These regulatory systems coordinate the spatial and temporal expression of key developmental genes, ultimately directing groups of undifferentiated cells to adopt specific organ identities. In the model plant *Arabidopsis thaliana*, these organs are arranged in four concentric whorls, each giving rise to a distinct floral structure [11].

At the centre of the *Arabidopsis* flower lies the female reproductive organ, known as the gynoecium. This structure plays a pivotal role in ensuring successful fertilisation of the ovules by pollen and subsequently develops into a fruit that supports embryo development and seed maturation [12]. A critical feature for the efficiency of this process is the radial symmetry of the gynoecium’s apical end, the style. In wild-type *Arabidopsis*, the style forms as a radially symmetrical, cylindrical structure atop the fused carpels that constitute the bilaterally symmetrical ovary [13]. This radial symmetry is essential for the uniform reception and growth of pollen tubes, which in turn ensures successful fertilisation. This fundamental process is controlled, among others, by auxin and CK [14].

Given the critical role of hormonal signalling in coordinating flower development, particularly the interplay between auxin and CK, understanding the transcriptional regulators that modulate the development of flower organs is essential. Among these, the adaxial-specific HD-ZIP II transcription factors *Homeobox Arabidopsis thaliana 3* (*HAT3*) and *Arabidopsis thaliana HomeoBox4* (*ATHB4*) have emerged as key regulators of multiple developmental processes in plants [15]. Originally characterised for their roles in shade avoidance, embryogenesis, and leaf and root development, HAT3 and ATHB4 have also been shown to influence the development of reproductive structures, including the gynoecium [16,17,18,19].

Recent studies have highlighted the role of HAT3 and ATHB4 in maintaining hormonal homeostasis during the bilateral-to-radial symmetry transition occurring during gynoecium development, particularly through the modulation of light, auxin, and CK pathways [17,20]. In *Arabidopsis* embryos and gynoecia, these transcription factors regulate auxin distribution and accumulation, which is crucial for establishing proper organ patterning and polarity [17,19]. In a double loss-of-function mutant, *hat3 athb4*, the gynoecium exhibits marked morphological alterations, especially at the apical tip, due to disrupted auxin dynamics. Aberrant patterns of the auxin-responsive reporter *DR5::GFP* [21], which shows precocious and elevated auxin signalling in medial apical tissues and a lack of signal in regions typically associated with adaxial identity, lead to the formation of a split-style phenotype, which shows bilateral symmetry [17].

This misregulation of auxin distribution in the *hat3 athb4* gynoecium can be partially rescued by the local application of NPA, an inhibitor of polar auxin transport [22], indicating that HAT3 and ATHB4 may regulate the expression of genes involved in auxin biosynthesis, degradation, or transport. Importantly, this aligns with the broader notion that organ shape and polarity, including the radial symmetry of the style, depend on finely tuned auxin biosynthesis and fluxes [23].

Moreover, HAT3 and ATHB4 also appear to function as repressors of CK output during reproductive development. In *hat3 athb4* mutants, exogenous CK application results in exaggerated proliferation of apical gynoecial tissues and the ectopic formation of trichomes on the valves, producing leaf-like features not observed in wild-type gynoecia [17]. This phenotype is consistent with enhanced CK responsiveness, as increased trichome density and complexity are commonly associated with elevated CK levels or signalling sensitivity [6,24].

These observations led us to hypothesise that HAT3 and ATHB4 serve as central regulators of the auxin–CK balance during flower development. Accordingly, beyond their established role in the gynoecium, we found that *hat3 athb4* mutants also show heightened sensitivity to CK in other floral organs, such as the sepals, again manifesting in altered trichome development. Furthermore, we show that the *hat3 athb4* mutant’s inflorescence exhibits an altered phyllotactic pattern, defined as the spatial arrangement of organs around the stem, a phenotype tightly regulated by auxin [3].

To elucidate the molecular basis of these phenotypes and identify the downstream targets of HAT3 and ATHB4, we performed RNA-seq analysis comparing inflorescences from *hat3 athb4* mutants and wild-type plants. Our transcriptomic data revealed that HAT3 and ATHB4 act upstream of a suite of genes involved in both CK and auxin metabolism and signalling. Specifically, they potentially regulate the expression of genes involved in CK degradation (CKXs) [25] and transport (PUPs) [26,27], as well as genes implicated in auxin biosynthesis (e.g., members of the IPyA and IAOx pathways) [28,29], transport (PID) [30], and signalling (ARF11) [31]. These findings support a model in which HAT3 and ATHB4 coordinate the spatial and temporal balance of phytohormones in floral tissues, thereby ensuring proper organ development and reproductive success.

## 2. Results

### 2.1. Loss of HAT3 and ATHB4 Leads to Cytokinin Hypersensitivity and Enhanced Trichome Complexity in Sepals

To investigate whether HAT3 and ATHB4 regulate trichome production through CK signalling, we treated flowers of both *hat3 athb4* double mutants and wild-type (WT) plants with benzyladenine (BA, 100 µM) and analysed the density and morphology of the trichomes on sepals (Figure 1).

Under mock (−BA) conditions, *hat3 athb4* sepals exhibited a significantly higher number of trichomes compared to WT sepals, with an average of approximately 5.5 trichomes per sepal in the mutant versus 1 in the WT (Figure 1A,B). In addition to increased density, a notable proportion of trichomes in the double mutant displayed a branched morphology (39.2%), in contrast to WT trichomes, which were predominantly unicellular and unbranched (Figure 1A,B).

In agreement with altered trichome development in the *hat3 athb4* sepals, we performed a gene network analysis using the GeneMANIA plugin in Cytoscape [32]. This analysis allowed us to predict functionally related genes by integrating co-expression, co-localization, and orthology prediction networks. This in silico analysis showed that HAT3 and ATHB4 are involved in developmental processes, including trichome morphogenesis (Appendix A), identifying interesting genes potentially regulated by HAT3 and ATHB4, such as members of the homeobox-leucine zipper family protein belonging to the HD-ZIP IV family and involved in trichome differentiation, such as HDG2, HDG11, and HDG12 (Appendix A) [33,34,35].

Upon CK treatment (+BA), the *hat3 athb4* sepals responded with a further increase in both trichome density and the proportion of branched trichomes, whereas WT sepals showed a more modest response (Figure 1A,B). These findings indicate that *hat3 athb4* sepals are hypersensitive to exogenous CK, mirroring the enhanced CK responsiveness previously observed in their gynoecia [17].

Next, we analysed two independent estradiol-inducible overexpressing lines on HAT3 (*XVE::HAT3*) [19]. We observed a reduction in the number of trichomes in the presence of CK, when *HAT3* overexpression was induced by estradiol (Figure 1C,D). Moreover, the presence of CK-induced branched trichomes was also reduced in the overexpressing lines treated with estradiol and CK (Figure 1C,D). These data show that HAT3-overexpressing lines are resistant to CK applications. Furthermore, they suggest that HAT3 and ATHB4 may act at the level of catabolism (e.g., increasing CK degradation), signalling (attenuating CK responses), and/or transport (redistributing CK).

Altogether, our data indicate that HAT3 and ATHB4 modulate CK sensitivity and contribute to the maintenance of CK homeostasis during reproductive development, including trichome differentiation on floral organs such as sepals.

### 2.2. Loss of HAT3 and ATHB4 Leads to Altered Phyllotaxy

The emergence of shoot apical organs, known as phyllotaxy, is a process regulated by auxin [3]. To determine whether HAT3 and ATHB4 are involved in the regulation of auxin during the establishment of inflorescence architecture, we measured the divergence angle at which flowers emerge laterally from the central axis. This analysis revealed a significant increase in the divergence angle of *hat3 athb4* flowers compared to the wild-type flowers (Figure 2). This phenotype suggests that HAT3 and ATHB4 may regulate auxin homeostasis, signalling, and/or transport, leading to misregulation of its patterning cues.

### 2.3. HAT3 and ATHB4 Transcriptionally Influence Hormonal Homeostasis

To investigate the molecular mechanisms by which HAT3 and ATHB4 modulate plant reproduction, we performed RNA-sequencing (RNA-seq) analysis on inflorescences of *hat3 athb4* and WT (Col-0) plants. This analysis revealed a substantial number of differentially expressed genes (DEGs), with 5361 upregulated and 5436 downregulated in the mutant background compared with the wild-type (Figure 3A and Appendix A). DEGs below an FDR cutoff of 0.5 were identified and the gene ontology (GO) enrichment analysis of the DEGs revealed significant overrepresentation of terms associated with hormonal regulation (Figure 3B,C and Appendix A). In particular, genes involved in the Gibberellin, Ethylene, CK, Brassinosteroid, auxin, and Abscisic Acid pathways were misregulated in the hat3 athb4 inflorescences (Figure 3D and Appendix A). This enrichment highlights the involvement of HAT3 and ATHB4 in regulating hormonal networks in reproductive tissues.

### 2.4. HAT3 and ATHB4 Transcriptionally Influence Auxin and Cytokinin Homeostasis

To investigate the molecular mechanisms by which HAT3 and ATHB4 modulate auxin and CK biology, which could explain the observed defects in trichome development and altered phyllotaxy (Figure 1 and Figure 2), we further interrogated our RNA-seq data.

In particular, genes involved in pathways linked to tryptophan metabolism, hormone response, hormone-mediated signalling, and auxin-specific signalling processes were either up- or downregulated in the *hat3 athb4* mutant background compared to WT (Figure 3B,C).

**Figure 3 plants-14-03723-f003:**
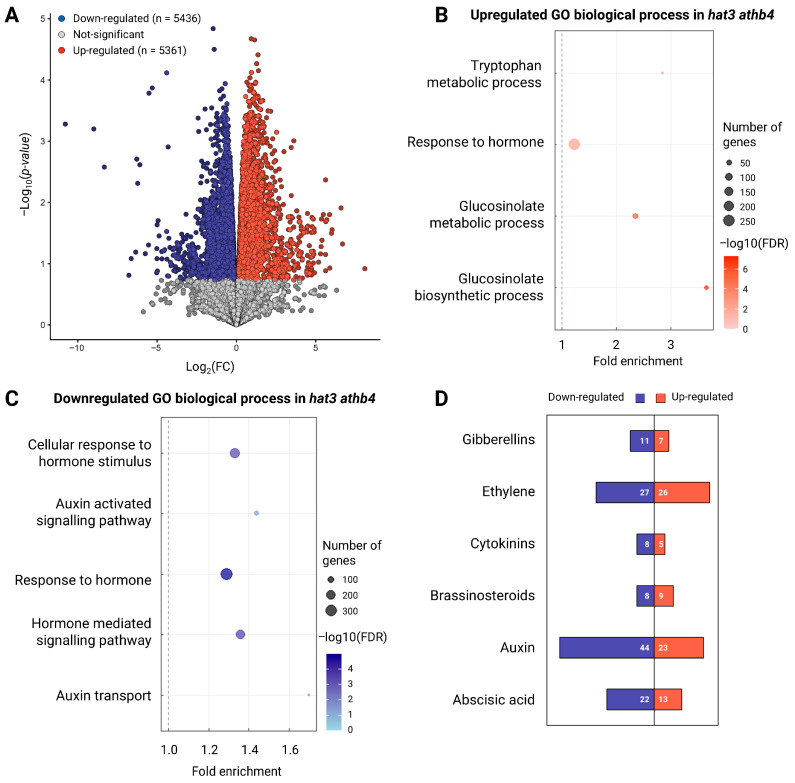
Overview of RNA-seq analyses. (**A**) Volcano plot showing the number of downregulated (blue) or upregulated (red) genes identified in the RNA-seq analyses. (**B**,**C**) Representative and significant biological process gene ontology terms associated with differentially expressed genes in *hat3 athb4* involved with hormone regulation, showing under-represented (**B**) and over-represented (**C**) biological processes in *hat3 athb4* relative to Col-0. Dot size indicates the number of differentially expressed genes associated with the process. Colour shade indicates the significance of the enrichment (−log10(FDR-corrected *p*-values)). The grey dashed line represents a fold enrichment of 1. (**D**) Number of downregulated (blue) and upregulated (red) genes associated with different hormones classes.

Focusing specifically on genes related to auxin and CK, our analysis revealed a clear transcriptional impact of HAT3 and ATHB4 on both hormone pathways. Notably, several genes involved in CK transport and degradation showed altered expression in the double mutant (Figure 4A). Among the hypothesised CK transporters, members of the *PURINE PERMEASE* (*PUP*) family were predominantly downregulated: four identified *PUPs* (*PUP1*, *PUP10*, *PUP14*, and *PUP21*) showed increased transcript levels in *hat3 athb4* inflorescences (Figure 4A). Although the precise roles of PUPs in intercellular CK transport remain to be fully elucidated, this general upregulation suggests that HAT3 and ATHB4 may act to control CK distribution within cells and/or tissues under normal developmental conditions.

In addition, our data revealed a change in the expression of *CYTOKININ OXIDASE*/*DEHYDROGENASE* (*CKX*) genes, which are involved in CK degradation. Both *CKX6* and *CKX3* were downregulated in the double mutant (Figure 4A).

We confirmed the roles of HAT3 and ATHB4 in CK degradation by analysing *CKX3* gene expression in mutant and wild-type inflorescences using qRT-PCR experiments. Our data showed that the expression of *CKX3* was downregulated in the *hat3 athb4* double mutant (Figure 4B), consistent with the observed hypersensitivity of the mutant to CK applications during trichome development on sepals (Figure 1A,B). Among the CK transporters, we analysed the expression of *PUP14* and confirmed the upregulation of its expression in the double mutant’s background (Figure 4B). Furthermore, our RNA-seq experiments also showed that *HDG11* was misregulated in the *hat3 athb4* double mutant, supporting the in silico analysis performed by using GeneMANIA (version 3.5.3) (Figure 4A, Appendix A).

Collectively, these findings provide evidence that HAT3 and ATHB4 are transcriptional regulators of multiple genes involved in the CK transport and degradation, as well as in the control of trichomes differentiation, establishing a complex role for these transcription factors in fine-tuning hormonal dynamics during flower development.

Next, we focused on the analysis of auxin-related genes in the inflorescences of *hat3 athb4* mutants compared to the WT. Our RNA-seq revealed that these transcription factors regulate auxin transport, biosynthesis, and signalling (Figure 5 and Appendix A). Auxin transport, in *hat3 athb4,* appears to be affected by the transcriptional upregulation of *PINOID* (*PID*) (Figure 5A), a key kinase that modulates the plasma membrane distribution of the PIN family’s auxin transporters [30]. Meanwhile, auxin signalling may be impacted via the upregulation of *AUXIN RESPONSE FACTOR 11* (*ARF11*) (Figure 5A).

Most notably, a substantial number of genes involved in auxin biosynthesis were differentially expressed, encompassing two major metabolic branches: the indole-3-acetaldoxime (IAOx) pathway and the indole-3-pyruvate (IPyA) pathway [29,36,37,38] (Figure 5A,B).

In the IPyA pathway, the TRYPTOPHAN AMINOTRANSFERASE OF ARABIDOPSIS (TAA1/TAR) gene family catalyses the conversion of tryptophan (TRP) to IPyA [36,39]. IPyA is then transformed into indole-3-acetic acid (IAA) through the action of flavin-containing monooxygenases encoded by the *YUCCA* (*YUC*) gene family [36,40,41]. *Arabidopsis thaliana* possesses 11 *YUC* genes, and their overexpression has been shown to result in strong auxin-overproduction phenotypes [42,43,44,45,46]. Conversely, *yuc* loss-of-function mutants exhibit auxin-deficiency symptoms, which can be partially restored through exogenous auxin supplementation or endogenous tissue-specific auxin production [44,47,48,49,50]. In our dataset, *YUC6* was downregulated in the double mutant (Figure 5A). To assess changes in *YUCCA* gene expression, we analysed the transcript levels of *YUC6* and additional family members by qRT-PCR. Although *YUC6* and *YUC8* expression was reduced in the double mutant, the difference was not statistically significant relative to the wild type. In contrast, *YUC5* and *YUC3* exhibited a significant decrease in expression. The data support a model in which HAT3 and ATHB4 positively regulate local auxin biosynthesis within reproductive tissues by activating the IPyA pathway (Figure 5B).

On the other hand, the IAOx pathway operates at a metabolic junction where IAOx can be diverted either towards IAA biosynthesis or towards the synthesis of indole glucosinolates (GLSs) (Figure 5B) [51]. This branch point is critical: impairment of the GLS pathway has been linked to elevated auxin levels, and vice versa, indicating a tightly regulated metabolic balance [51,52].

Our RNA-seq data revealed the upregulation of several genes at the crux between IAA and GLS biosynthesis (Figure 5B). *CYP79B2*, a gene encoding an enzyme that catalyses the initial step in the conversion of TRP to IAOx, which is a precursor common to both the IAA and GLS biosynthetic branches [53], was found to be upregulated in the double mutant (Figure 5A). Notably, plants overexpressing *CYP79B2* exhibit auxin overproduction traits, whereas double mutants of *cyp79B2* and *cyp79B3* show reduced IAA content and phenotypes consistent with impaired auxin biosynthesis [54]. Accordingly, qRT-PCR experiments revealed that the expression of *CYP79B2* was upregulated in the double mutant’s background (Figure 5C). Therefore, the results of our analysis suggest that *hat3 athb4* mutants may accumulate increased IAOx levels.

Moreover, several other genes involved in diverting IAOx towards GLS production were found to be upregulated in the *hat3 athb4* background: these are *SUPERROOT1* (*SUR1*) and *SUPERROOT2* (*SUR2*), whose proteins convert IAOx in thiohydroximates and then to indolyl GLS (Figure 5A,B) [54,55,56,57]. Our qRT-PCR experiments showed that the expression of *SUR1*, but not *SUR2*, was statistically upregulated in the double mutant’s background (Figure 5C). Furthermore, the misregulation of genes involved in indolyl GLS turnover was also observed. Specifically, *THIOGLUCOSIDE GLUCOHYDROLASE 1* (*TGG1*), *TGG2*, *EPITHIOSPECIFIER PROTEIN* (*ESP*) and *EPITHIOSPECIFIER MODIFIER 1* (*ESM1*) catalyse myrosinase enzymes, which affects the production of indole-3-acetonitrile (IAN) and can be hydrolysed by nitrilases (such as *NIT1*) back to IAA (Figure 5B), and were found upregulated in the double mutant (Figure 5A,B). All these enzymes are critical in titrating IAA and GLS production, contributing to a trade-off between plant growth and defence [51,58].

Additionally, the gene encoding the enzyme amidase1 (AMI1) that converts indole-3-acetamide (IAM) into IAA (Figure 5B) was found to be upregulated in the *hat3 athb4* mutant background in both our RNA-seq and qRT-PCR experiments (Figure 5A,C).

Collectively, these results point to a role for HAT3 and ATHB4 in maintaining hormonal homeostasis in developing inflorescences. These transcription factors seem to promote auxin biosynthesis via the IPyA/YUC pathway, although whether this regulation is direct or indirect remains to be clarified. Moreover, our data suggest that HAT3 and ATHB4 may also modulate GLS biosynthesis by titrating auxin production through the IAOx route. Their loss results in an imbalance that might shift precursor allocation, diminishing auxin production, which may have downstream consequences for floral patterning and development.

## 3. Discussion

In this study, we demonstrate that two HD-ZIP class II transcription factors, HAT3 and ATHB4, play crucial roles in modulating hormonal balance during flower development. Our phenotypical and molecular analyses show that *hat3 athb4* double mutants produce ectopic and branched trichomes on sepals and display an altered phyllotaxy. These phenotypes, together with the previously reported disruption of gynoecium development [17,20], represent hallmarks of disturbed hormone homeostasis, particularly involving CK and auxin.

Our RNA-seq analysis reveals that HAT3 and ATHB4 regulate key components of both the auxin and CK pathways. Specifically, they appear to promote auxin biosynthesis, signalling, and transport (Figure 5) while modulating CK transport and degradation (Figure 4). These findings suggest that HAT3 and ATHB4 are pivotal in fine-tuning the auxin–CK balance in floral tissues, thereby ensuring correct organ specification and tissue identity. Our RNA-seq analysis highlighted downregulation of the *CKX6* gene (Figure 4A). Consistently with the role of the HAT3 and ATHB4 transcription factors in the shade-avoidance response, CKX6 had been previously reported to be part of a regulatory circuit in canopy shade [59]. *CKX6* was shown to be induced in low R/FR light [60], and it was found to promote auxin-induced CK breakdown in leaf primordia under low R/FR light, thereby reducing leaf cell proliferation in shade [61]. These findings are in line with our transcriptional analysis showing that another cytokinin oxidase, *CKX3*, was found to be downregulated in the *hat3 athb4* background (Figure 4A,B). Moreover, they point to a broader role for HAT3 and ATHB4 as mediators between environmental cues and hormonal signals. Furthermore, the hypersensitivity of *hat3 athb4* tissues to exogenous CK, showed by the higher trichome density and complexity (Figure 1A,B), supports the scenario in which these transcription factors normally function to dampen CK response, potentially by restricting its accumulation or signal output. It remains unclear whether the augmented differentiation of trichomes in the double mutant, and/or the hypersensitivity to CK, work via HDG11, as suggested by our in silico analysis (Appendix A) and the results from our RNA-seq (Figure 4A).

Auxin has long been recognised for its role as a developmental cue during organogenesis. Functioning as a morphogen, auxin gradients define positional information, influencing cell division, differentiation, and tissue patterning in a concentration-dependent manner [62,63]. Its importance during flower development, particularly in the gynoecium and associated reproductive structures, is well established [23,64]. The *hat3 athb4* double mutants show several auxin-related phenotypes that have been previously described, such as asymmetrical ovaries [17,20], monocotyledonous seedlings, inactive shoot apical meristems, altered leaf shape and polarity, defective vascular patterning [19], and root apical meristem development [65]. These phenotypes align with perturbed auxin biosynthesis, signalling, and transport. Our RNA-seq suggest that potentially every part of auxin biology is misregulated in the double mutant, which showed altered expression of *ARF11*, *PID*, and several genes involved in its biosynthesis (Figure 5).

However, auxin does not act on its own. It interacts dynamically with other hormonal pathways, such as CK, as well as with metabolic networks, contributing to a robust but highly responsive developmental framework.

Another intriguing finding from our transcriptomic data is that HAT3 and ATHB4 may also potentially act as a molecular bridge between auxin and indolic GLS biosynthesis pathways. GLS are Brassicaceae-specific secondary metabolites with known defensive, flavour, and health-related properties. While their defensive roles are well-documented, their regulatory integration with hormone signalling, particularly auxin, remains poorly understood [51,66]. Our preliminary evidence suggests that HAT3 and ATHB4 not only control auxin biosynthesis but may also regulate genes in the GLS pathway, although metabolic measurements of endogenous GLS content will be required to confirm this possibility. Speculatively, this hints at a potential trade-off between growth/development and defence, a well-known yet complex relationship in plant biology. If supported by further quantitative and genetic data, it would be plausible to propose a scenario in which HAT3 and ATHB4 contribute to the balancing act between resource investment in growth versus defence by modulating both hormonal and metabolic gene networks in leaves as well as in the inflorescences.

Taken together, our findings position HAT3 and ATHB4 as central nodes in a complex network of developmental regulation, bridging hormonal signalling with secondary metabolism. Their role in controlling auxin and CK activity is particularly critical in reproductive tissues, where precise spatial and temporal coordination is essential for successful fertilisation and seed development. The potential link to GLS metabolism further expands the developmental relevance of these transcription factors and suggests broader roles in coordinating developmental and defence-related pathways.

Future studies should aim to dissect the tissue-specific regulatory mechanisms by which HAT3 and ATHB4 influence their downstream targets, as well as to explore how their activity integrates environmental cues, such as light and stress, into developmental outputs. Understanding the molecular links behind these interactions will not only enhance our fundamental knowledge of plant development but may also inform new strategies for crop improvement. Flowers are the cornerstone of sexual reproduction in angiosperms and are indispensable to human agriculture. They enable the propagation of crop species and the production of fruits and grains central to our food systems. In an era marked by climate instability and population growth, improving our understanding of floral development is more than a matter of scientific curiosity; it is a prerequisite for innovation in crop engineering and sustainable food production. Insight into the regulatory mechanisms guiding flower formation could facilitate precise manipulation of reproductive traits, ultimately contributing to efforts aimed at securing global food supply. For instance, targeted manipulation of HAT3/ATHB4 or their downstream pathways could enable the engineering of floral traits optimised for yield, resilience, or nutritional quality, particularly in Brassicaceae crops.

## 4. Materials and Methods

### 4.1. Plant Growth Conditions

The plant material used was *Arabidopsis thaliana* in the wild-type Col-0 ecotype background. *hat3 athb4*, a loss-of-function insertional line, and the inducible *XVE::HAT3* overexpressing lines were developed by [19]. Plants were grown on Levington F2 soil under standard long-day conditions (16 h light/8 h dark) in controlled environment rooms.

### 4.2. RNA Extraction and Sequencing

RNA was extracted from inflorescences of Col-0 and *hat3 athb4* plants. The material was collected between 10 and 11 am (to maintain circadian consistency) with three biological replicates per genotype at 3–5 weeks post-germination. Extraction was performed using the RNeasy Plant Mini Kit (Qiagen, Venlo, Netherlands) following the manufacturer’s protocol. RNA samples were shipped on dry ice to GENEWIZ for sequencing. Raw sequencing reads were quantified against the *Arabidopsis thaliana* reference transcriptome (TAIR CDS fasta) using Kallisto. Gene ontologies (GOs) were identified from the differentially expressed genes (DEGs) with an arbitrary false discovery rate (FDR) of 0.5 to balance false positives and gene retention. Within this threshold, a further false discovery rate (FDR) *p* < 0.05 was applied within Panther to obtain an overview of trends in the data. Plots were generated using the *ggplot2* R package (version 4.0.0), following established protocols [67]. Heatmaps visualising differential gene expressions were generated and Z-scores were calculated using the normalised counts per million (CPM) data for each gene using the *ggplot2* R package.

### 4.3. Reverse Transcription Quantitative Polymerase Chain Reaction (RT qPCR)

Total RNA from flowers of Col-0 and *hat3 athb4* was isolated as described above. cDNA synthesis was performed using M-MLV Reverse Transcriptase (Promega, Madison, Wisconsin, USA) according to the manufacturer’s instructions. qRT-PCR experiments were carried out on a BioRad CFX96 system using gene-specific primers (see below) with three biological replicates per background and four technical replicates per sample using SYGREEN BLUE qPCR MIX (PCRBIO, London, UK). Expression levels were normalised to UBIQUITIN 10 and ACTIN2 (used as the housekeeping genes) and calculated using the 2^−ΔΔCt^ method.

### 4.4. Primers for qRT-PCR

The primers are listed in Appendix A [26,68,69,70,71,72,73].

### 4.5. Cytokinin and β-Estradiol Treatments

Inflorescences of each background, *hat3 athb4* and Col-0, were treated upon bolting with 100 µM 6-Benzylaminopurine (BA) (Merck, Darmstadt, Germany) and 0.01% Silwet-77. Mock-treated plants were only sprayed with a mock solution containing NaOH and 0.01% Silwet-77. Plants were sprayed either with mock or CK every 4 days, 4 times total, and flower samples were collected for phenotypic analysis 4 days after treatment.

Inflorescences of *XVE::HAT3* plants were sprayed once a day for 4 consecutive days with a mix of 100 μM BA and mock or with 100 μM BA and 20 μM β-estradiol, as previously described [17]. All spray treatments used a 0.01% final concentration of Silwet L-77. Trichomes on sepals were counted under a stereomicroscope after 4 days.

Statistical analysis was performed using the GraphPad QuickCalcs free online page (https://www.graphpad.com/quickcalcs/ accessed on 14–15 November 2025).

### 4.6. Scanning Electron Microscopy

Col-0 and *hat3 athb4* flower samples were collected and fixed in a prepared solution of Formaldehyde, Alcohol, and Acetic Acid (FAA) for four hours prior to a series of ethanol (EtOH) dilutions to 100% dry EtOH, prior to reaching the critical point of drying (CPD) using a Leica CPD300 (Leica Microsystems, Wetzlar, Germany). The dried samples were then dissected at the light microscope and placed on stubs prior to sputter coating using Au (gold) at 7.5 nm, using a Leica 600 ACE sputter coater (Leica Microsystems, Wetzlar, Germany). The samples were imaged using the FEI Nova NanoSEM (FEI, Hillsboro, OR, USA).

### 4.7. Phyllotaxy Measurements

The divergence angle of mature, non-senescing siliques was measured using plants approximately 3 weeks old. Only the main stem was measured by eye using a protractor. The circular (360°) protractor was held over a stand where the main stem was positioned upright and the clockwise divergence angle was measured between successive siliques, measuring from the point of emergence of the siliques from the stem. This was completed along the length of the stem for 17 plants for *hat3 athb4* and 11 plants for Col-0.

## Figures and Tables

**Figure 1 plants-14-03723-f001:**
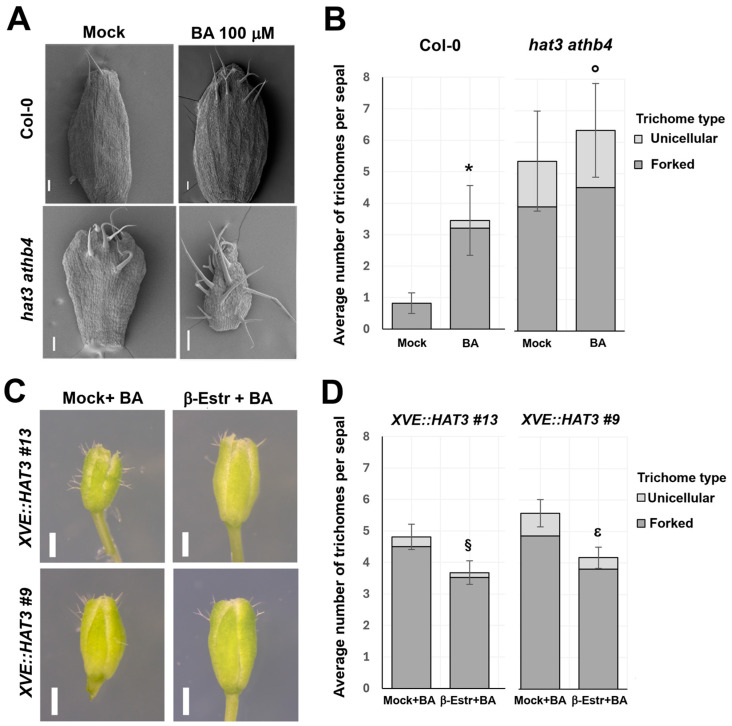
HAT3 and ATHB4 control CK-mediated trichome development on sepals. (**A**) Representative SEM images of *hat3 athb4* and wild-type (WT, Col-0) sepals, treated with 100 µM CK (+BA) or mock (−BA). Scale bars represent 100 µm. (**B**) Quantification of number of trichomes per sepal in Col-0 and *hat3 athb4* treated as in (**A**). Quantification of forked trichomes (light-grey bars) and unicellular trichomes (dark-grey bars) is shown. (**C**) Stereo micrographs of two *XVE::HAT3* lines treated with 100 μM BA and either mock or 20 μM β-estradiol to induce the overexpression of *HAT3*. (**D**) Quantification of number of trichomes per sepal of the two *XVE::HAT3* lines and treatments showed in (**C**). Forked trichomes and unicellular trichomes are shown. The number of trichomes on sepals were compared using *t*-test analysis. Two-tailed *p* values are as follows: BA vs. mock in Col-0, *p* < 0.0001 (*); *hat3 athb4* vs. Col-0 *p* < 0.0001 (°); β-Estr + BA vs. β-Estr + mock in *XVE::HAT3 #13*, *p* = 0.0374 (§); β-Estr + BA vs. β-Estr + mock in *XVE::HAT3 #9*, *p* = 0.0074 (ε). *p* values < 0.05 were considered statistically significant. *p* values < 0.001 were considered extremely statistically significant. n ≥ 129 sepals (collected from multiple flowers of the same plant as well as from different plants, treated as independent biological replicates).

**Figure 2 plants-14-03723-f002:**
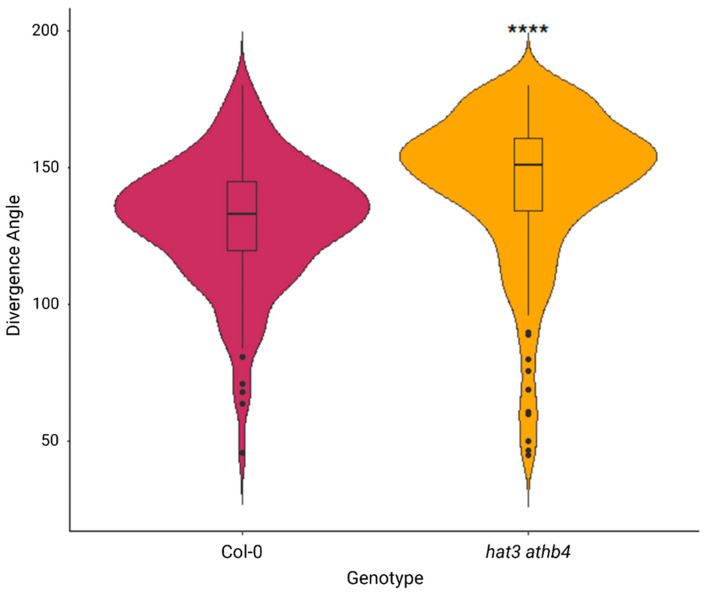
Phyllotactic divergence angles between wild-type and *hat3 athb4* emerged siliques. Significance was measured using Welch’s two tailed *t*-test, *p* < 0.0001 (****). n = 11 plants for WT (Col-0), with 132 divergence angles measured; n = 17 plants for *hat3 athb4* with 172 divergence angles measured.

**Figure 4 plants-14-03723-f004:**
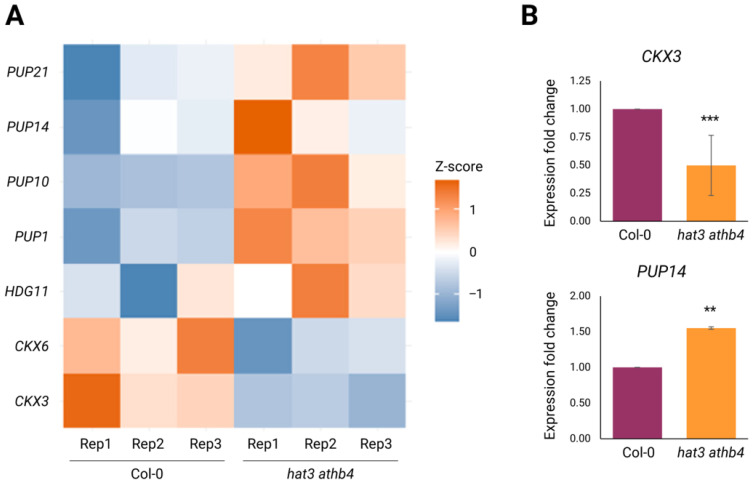
Regulation of CK biosynthesis and transport by HAT3 and ATHB4. (**A**) Heatmap showing differentially expressed genes in *hat3 athb4* identified by RNA-seq, related to CK biology. Z-score scale bar from −1 (downregulation in blue) to +1 (upregulation in red). Rep, biological replicate. (**B**) qRT-PCR results of *CKX3* and *PUP14* expression showing significant modulation in *hat3 athb4* compared to WT (Col-0). Significance was calculated using Student’s two-tailed *t*-test and was plotted in R (**, *p* < 0.01; ***, *p* < 0.001), N ≥ 2.

**Figure 5 plants-14-03723-f005:**
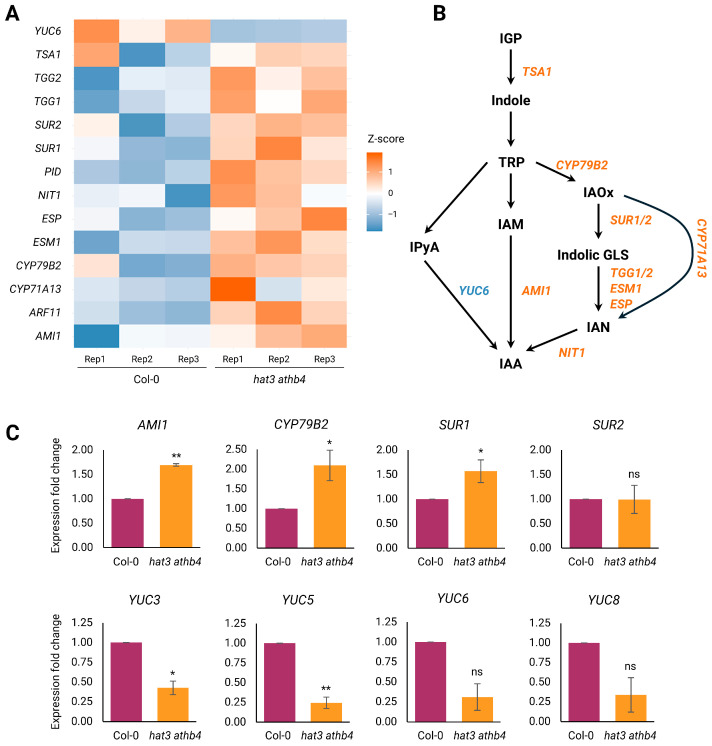
Regulation of auxin biosynthesis and transport by HAT3 and ATHB4. (**A**) Heatmap showing differentially expressed genes in *hat3 athb4* identified by RNA-seq, related to auxin biology. Z-score scale bar from −1 (downregulation, in blue) to +1 (upregulation, in red). Rep, biological replicate. (**B**) Diagram of IAA biosynthetic pathways. Biosynthetic enzymes highlighted in blue and red were found in the RNA-seq analysis in (**A**) and were downregulated and upregulated, respectively, in *hat3 athb4* compared to WT (Col-0). (**C**) qRT-PCR results of the expression of genes involved in IAA biosynthetic pathways in *hat3 athb4* compared to WT (Col-0). Significance was calculated using Student’s two-tailed *t*-test and plotted in R (*, *p* < 0.05; **, *p* < 0.01; ns, not significant), N ≥ 2.

## Data Availability

The raw RNA-seq data from *hat3 athb4* and Col-0 inflorescences will be made available upon request.

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
