# Peer review of "The HD-ZIP II Transcription Factors HAT3 and ATHB4 Fine-Tune Auxin and Cytokinin Pathways During Flower Development"

_plants, 2025, doi:10.3390/plants14243723_

Round 1

Reviewer 1 Report

Comments and Suggestions for Authors

This manuscript provides a clear and well-structured set of data on the role of HAT3 and ATHB4 in coordinating auxin and cytokinin pathways during flower development. The experiments are well executed, and the integration of the phenotypic observations with the transcriptomic analysis adds value to the study. Overall, the work is solid and contributes useful insight into hormonal balance during reproductive development.

I have only a few minor points that I believe would strengthen the manuscript further:

RNA-seq reporting
It would be helpful to provide a little more detail on the RNA-seq analysis so readers can better assess the robustness of the differential expression results. In particular, stating the criteria used to define DEGs (e.g., FDR threshold and any log₂FC cutoff), the total number of up- and down-regulated genes, and a brief note on replicate consistency or alignment metrics would make this section more complete.

IAOx/GLS interpretation
The idea that HAT3/ATHB4 may influence the balance between IAA and glucosinolates at the IAOx branch point is interesting, but currently relies on transcript changes alone. As no metabolite measurements are shown, I would suggest presenting this more as a hypothesis supported by the RNA-seq trends, rather than as a firm conclusion.

Auxin biosynthesis wording
The statement that HAT3/ATHB4 “promote auxin biosynthesis via the IPA/YUC pathway” could be interpreted as a direct regulatory effect. Since the evidence is based on transcript levels (e.g., reduced YUC6 expression), a slightly softer wording may better reflect that the data are consistent with reduced local auxin biosynthesis in the mutant, without implying direct regulation.

Quantification in figures
For consistency, it would be useful if all quantified figure panels clearly state the statistical test used, the sample size (n), and whether the values represent individual organs or individual plants. This information is present for Fig. 2, and adding it uniformly across all figures would avoid any ambiguity.

The manuscript is strong, and these are small refinements aimed at improving clarity. I believe the study will be suitable for publication once these points are addressed.

Author Response

Reviewer 1:

This manuscript provides a clear and well-structured set of data on the role of HAT3 and ATHB4 in coordinating auxin and cytokinin pathways during flower development. The experiments are well executed, and the integration of the phenotypic observations with the transcriptomic analysis adds value to the study. Overall, the work is solid and contributes useful insight into hormonal balance during reproductive development.

I have only a few minor points that I believe would strengthen the manuscript further:

1) RNA-seq reporting
It would be helpful to provide a little more detail on the RNA-seq analysis so readers can better assess the robustness of the differential expression results. In particular, stating the criteria used to define DEGs (e.g., FDR threshold and any log₂FC cutoff), the total number of up- and down-regulated genes, and a brief note on replicate consistency or alignment metrics would make this section more complete.

Response: To provide more detail on the RNA-seq analysis we revised the manuscript as follows:

- In the Results section (2.3. HAT3 and ATHB4 Transcriptionally Influence Hormonal Homeostasis) we specified that “Differentially expressed genes (DEGs) below an FDR cutoff of 0.5 were identified”.

- We have produced a revised version of Figure 3, in which panel A now displays a volcano plot showing the number of genes up- and down-regulated in the mutant background compared to the wild type, which are 5,361 and 5,436, respectively. These data have been also added to the Results section, in a new paragraph “2.3. HAT3 and ATHB4 Transcriptionally Influence Hormonal Homeostasis”.

- We added the following information to the Material & Methods (4.2. RNA Extraction and Sequencing): “Gene Ontologies (GOs) were identified from the differentially expressed genes (DEGs) with the arbitrary false discovery rate (FDR) of 0.5 to balance false positives and gene retention. Within this threshold, a further false discovery rate (FDR) P < 0.05 was applied within Panther to get an overview of trends in the data. Plots were generated using ggplot2 R package following established protocols [66]. Heat-maps visualising differential gene expression were generated and Z-scores were calculated using the normalised counts per million (CPM) data for each gene using ggplot2 R package.”

2) IAOx/GLS interpretation
The idea that HAT3/ATHB4 may influence the balance between IAA and glucosinolates at the IAOx branch point is interesting, but currently relies on transcript changes alone. As no metabolite measurements are shown, I would suggest presenting this more as a hypothesis supported by the RNA-seq trends, rather than as a firm conclusion.

Response: We have modified the text as follows:

- Results: “Moreover, our data suggest that HAT3 and ATHB4 may also modulate GLS biosynthesis by titrating auxin production through the IAOx route.”

- Discussion: “Our preliminary evidence suggests that HAT3 and ATHB4 not only control auxin biosynthesis but may also regulate genes in the glucosinolate pathway, although metabolic measurements of endogenous glucosinolates content will be required to confirm this possibility. Speculatively, this hints at a potential trade-off between growth/development and defense, a well-known yet complex relationship in plant biology. If supported by further quantitative and genetic data, it would be plausible to propose a scenario in which HAT3 and ATHB4 contribute to the balancing act between resource investment in growth versus defense by modulating both hormonal and metabolic gene networks in leaves as well as in the inflorescences.”

3) Auxin biosynthesis wording
The statement that HAT3/ATHB4 “promote auxin biosynthesis via the IPA/YUC pathway” could be interpreted as a direct regulatory effect. Since the evidence is based on transcript levels (e.g., reduced YUC6 expression), a slightly softer wording may better reflect that the data are consistent with reduced local auxin biosynthesis in the mutant, without implying direct regulation

Response: We have modified the conclusions of the Results section as follows:

 “These transcription factors seem to promote auxin biosynthesis via the IPyA/YUC pathway, although whether this regulation is direct or indirect remains to be clarified.”

Quantification in figures
For consistency, it would be useful if all quantified figure panels clearly state the statistical test used, the sample size (n), and whether the values represent individual organs or individual plants. This information is present for Fig. 2, and adding it uniformly across all figures would avoid any ambiguity.

Response: We have added the missing information in the caption of figure 1, as follows: “The number of trichomes on sepals were compared using t-test analysis. Two-tailed P values are as follows: BA vs Mock in Col-0, P<0.0001 (*); hat3 athb4 vs Col-0 P < 0.0001 (°); b-Estr + BA vs b-Estr + Mock in XVE::HAT3 #13, P = 0.0374 (§); b-Estr + BA vs b-Estr + Mock in XVE::HAT3 #9, P = 0.0074 (e). P values < 0.05 were considered statistically significant. P values < 0.001 were considered extremely statistically significant. n ≥ 129 sepals (collected from multiple flowers of the same plant as well as from different plants, treated as independent biological replicates).”

The manuscript is strong, and these are small refinements aimed at improving clarity. I believe the study will be suitable for publication once these points are addressed.

Response: We thank the reviewer for their supportive statement.

Reviewer 2 Report

Comments and Suggestions for Authors

The authors of this article performed experiments in Arabidopsis and found phenotypic differences between wild-type and mutant plants (transcription factors HAT3 and ATHB4). In addition, the authors also performed transcriptome sequencing of wild-type and mutant plants, identifying a series of auxin related genes.

The content of this article is rich , and the experiments are substantial. We recommended acceptance after minor revisions. We have some suggestions for the author's reference.

  1. The author did not check the hormone content in the mutant. We recommended to check them.

  1. In Figure 4B, the author only performed RT-PCR for the CK gene. It is recommended to also check the expression levels of key genes in the auxin pathway (RT-PCR).

  1. In Figure 5, the transcriptome analysis only involves the auxin signaling pathway. It is recommended to analyze other hormone pathways.

  1. We suggest that the author would upload the transcriptome raw data of this study to the SRA database at NCBI for the convenience of peers.

Author Response

Reviewer 2:

The authors of this article performed experiments in Arabidopsis and found phenotypic differences between wild-type and mutant plants (transcription factors HAT3 and ATHB4). In addition, the authors also performed transcriptome sequencing of wild-type and mutant plants, identifying a series of auxin related genes.

The content of this article is rich , and the experiments are substantial. We recommended acceptance after minor revisions. We have some suggestions for the author's reference.

 Response: We thank the reviewer for their supportive statement.

  1. The author did not check the hormone content in the mutant. We recommended to check them.

Response: We agree with the reviewer that an important point arising from our transcriptomic analysis is the need to assess the actual hormonal contents in hat3 athb4 double-mutant inflorescences compared with the wild type. However, we believe that quantitative metabolic measurements coupled with genetic analyses constitute substantial findings, which we intend to present in a follow-up study. 

  1. In Figure 4B, the author only performed RT-PCR for the CK gene. It is recommended to also check the expression levels of key genes in the auxin pathway (RT-PCR).

 Response:  We have added the expression analysis of PUP14 (for CK) in Figure 4B and the following genes for IAA and GLS production in Figure 5C: CYP79B2, YUC3, YUC5, YUC6, YUC8, SUR1, SUR2, and AMI1.

  1. In Figure 5, the transcriptome analysis only involves the auxin signaling pathway. It is recommended to analyze other hormone pathways.

Response: We have produced a revised version of Figure 3, in which panel D now displays the genes belonging to various hormonal pathways (i.e. gibberellin, ethylene, cytokinins, brassinosteroids, auxin, and abscisic acid) that fall within the GO category “Response to hormone” and are up- or down-regulated in our analysis.

We have added a paragraph in the results section to report on these results:

2.3. HAT3 and ATHB4 Transcriptionally Influence Hormonal Homeostasis

To investigate the molecular mechanisms by which HAT3 and ATHB4 modulate plant reproduction, we performed RNA-sequencing (RNA-seq) analysis on inflorescences of hat3 athb4 and WT (Col-0) plants. This analysis revealed a substantial number of differentially expressed genes (DEGs), with 5,361 upregulated and 5,436 downregulated in the mutant background compared with the wild type (Figure 3A and Supplemental Table 2). DEGs below an FDR cutoff of 0.5 were identified and the Gene Ontology (GO) enrichment analysis of the DEGs revealed significant overrepresentation of terms associated with hormonal regulation (Figure 3B,C and Supplemental Table 2). In particular, genes involved in Gibberellin, Ethylene, CK, Brassinosteroid, Auxin and Abscisic Acid pathways were misregulated in the hat3 athb4 inflorescences (Figure 3D and Supplemental Table 2). This enrichment highlights the involvement of HAT3 and ATHB4 in regulating hormonal networks in reproductive tissues.

  1. We suggest that the author would upload the transcriptome raw data of this study to the SRA database at NCBI for the convenience of peers.

Response: We will upload the transcriptome raw data of this study to the NCBI database as soon as another work, currently ongoing, will be published.

Reviewer 3 Report

Comments and Suggestions for Authors

The manuscript by Maio et al. investigates the roles of HAT3 and ATHB4 in flower development through phenotypic observation and RNA-seq analysis. While the overall logic of the writing is acceptable, the interpretation and validation of the RNA-seq data remain insufficient and require further revision.

Major Concerns:

1) In Figures 1B and 1C, statistical analysis is lacking, and the sample numbers are not indicated. Additionally, phenotypic images for the XVE3::HAT3 line are missing.

2) In Figure 2, only statistical data are presented without corresponding phenotypic images, making it difficult for readers to understand which specific traits the data refer to.

3) In Figure 3, the use of a gray background impedes clear visualization. Furthermore, the RNA-seq data could be more comprehensively utilized. For reference, please consider: Potential Secretory Transporters and Biosynthetic Precursors of Biological Nitrification Inhibitor 1,9-Decanediol in Rice as Revealed by Transcriptome and Metabolome Analyses. Rice Science, 2024, 31(1): 87–102.

4) In Figure 4, the authors analyze cytokinin (CK)-related genes based on RNA-seq results but only validate CKX3 via qPCR. Since the purpose here is to verify RNA-seq reliability through qPCR, validating only CKX3 is insufficient. Additionally, biological replicates for brassinosteroid (BR) experiments should be relabeled to avoid ambiguity.

5) In Figure 5B, the authors suggest that TSB3 catalyzes the conversion from indole to tryptophan (Trp). Please provide supporting references, as existing literature indicates that TSB1/TSB2 are responsible for this step. Moreover, other genes involved in the conversion from IGP to indole should also be examined for expression changes. Reference: The biosynthesis of auxin: how many paths truly lead to IAA?. Plant Growth Regulation, 78(3): 275–285.

Additionally, IAMH1/2 are currently considered the primary catalysts for IAM-to-IAA conversion. Have these genes been analyzed for expression changes? Reference: Two homologous INDOLE-3-ACETAMIDE (IAM) HYDRALASE genes are required for the auxin effects of IAM in Arabidopsis. Journal of Genetics and Genomics, 2020, 47(3).

Finally, while the authors validated selected CK-related genes, similar validation should be performed for auxin-related genes discussed in this section.

Minor Concerns:

1) The keywords are excessive and inadequately selected; please revise them.

2) The abbreviation "CK" for cytokinin should be used consistently throughout the manuscript instead of alternating between full and abbreviated forms.

3) Gene symbols and mutant names should be italicized.

4) In the Introduction, “…members of the YUCCA family and IAOx pathways” should be corrected to “…members of the IPyA pathway and IAOx pathway.”

5) In the Results section, “indole-3-pyruvate (IPA)” is abbreviated as “IPyA” elsewhere. The authors should uniformly use “IPyA.”

6) In the Results section, the statement “Conversely, yuc loss-of-function mutants exhibit auxin-deficiency symptoms…” requires a supporting citation, as only yuc8, a single mutation of YUC, has been clearly demonstrated to display notable defects under cytokinin conditions. Please cite: Functional roles of Arabidopsis CKRC2/YUCCA8 gene and the involvement of PIF4 in the regulation of auxin biosynthesis by cytokinin. Scientific Reports, 6, 36866.

Author Response

Reviewer 3:

The manuscript by Maio et al. investigates the roles of HAT3 and ATHB4 in flower development through phenotypic observation and RNA-seq analysis. While the overall logic of the writing is acceptable, the interpretation and validation of the RNA-seq data remain insufficient and require further revision.

Major Concerns:

1) In Figures 1B and 1C, statistical analysis is lacking, and the sample numbers are not indicated. Additionally, phenotypic images for the XVE3::HAT3 line are missing.

Response: We have added the missing information making the following changes:

- in Figure 1, we added scale bars and statistical significance to revised panels B and D;

- in the caption of figure 1, we added the following information: “The number of trichomes on sepals were compared using t-test analysis. Two-tailed P values are as follows: BA vs Mock in Col-0, P<0.0001 (*); hat3 athb4 vs Col-0 P < 0.0001 (°); b-Estr + BA vs b-Estr + Mock in XVE::HAT3 #13, P = 0.0374 (§); b-Estr + BA vs b-Estr + Mock in XVE::HAT3 #9, P = 0.0074 (e). P values < 0.05 were considered statistically significant. P values < 0.001 were considered extremely statistically significant. n ≥ 129 sepals (collected from multiple flowers of the same plant as well as from different plants, treated as independent biological replicates).”

- in the Materials & Methods, section 4.5, we added the following information: “Trichomes on sepals were counted under the stereomicroscope. Statistical analysis was performed using the GraphPad QuickCalcs free on line page (https://www.graphpad.com/quickcalcs/).

Furthermore, we have added representative stereomicroscope images of the XVE:HAT3 lines in the revised Figure 1C.

2) In Figure 2, only statistical data are presented without corresponding phenotypic images, making it difficult for readers to understand which specific traits the data refer to.

Response: In the Results section 2.2. Loss of HAT3 and ATHB4 Leads to altered phyllotaxy, we explained that “we measured the divergence angle at which flowers emerge laterally from the central axis.” Also, we titled the caption of Figure 2 “Phyllotactic divergence angles between wild-type and hat3 athb4 emerged siliques.” to explain the trait analysed.

Moreover, we provided a seminal reference (Reinhardt, D.; Pesce, E.-R.; Stieger, P.; Mandel, T.; Baltensperger, K.; Bennett, M.; Traas, J.; Friml, J.; Kuhlemeier, C. Regulation of Phyllotaxis by Polar Auxin Transport. Nature 2003, 426, 255–260, doi:10.1038/nature02081) that explains the phenotype analysed.

3) In Figure 3, the use of a gray background impedes clear visualization. Furthermore, the RNA-seq data could be more comprehensively utilized. For reference, please consider: Potential Secretory Transporters and Biosynthetic Precursors of Biological Nitrification Inhibitor 1,9-Decanediol in Rice as Revealed by Transcriptome and Metabolome Analyses. Rice Science, 2024, 31(1): 87–102.

Response: As suggested by the reviewer, we have revised Figure 3 to improve the presentation and visualization of our RNA-seq analysis. Specifically, in panel A we now display a volcano plot showing the number of up- and down-regulated genes; in panels B and C, we use colour gradients (red and blue) to indicate GO categories that are up- or down-regulated, respectively; and in panel D, we show the genes belonging to various hormonal pathways (i.e., gibberellins, ethylene, cytokinins, brassinosteroids, auxin, and abscisic acid) that fall within the GO category “response to hormone” and are up- or down-regulated in our dataset.

4) In Figure 4, the authors analyze cytokinin (CK)-related genes based on RNA-seq results but only validate CKX3 via qPCR. Since the purpose here is to verify RNA-seq reliability through qPCR, validating only CKX3 is insufficient. Additionally, biological replicates for brassinosteroid (BR) experiments should be relabeled to avoid ambiguity.

Response: In addition to CKX3, the revised Figure 4 now also presents data for PUP14.

In addition, to indicate Biological Replicates we have changed the abbreviation from BR1,2,3 to Rep1,2,3.

5) In Figure 5B, the authors suggest that TSB3 catalyzes the conversion from indole to tryptophan (Trp). Please provide supporting references, as existing literature indicates that TSB1/TSB2 are responsible for this step.

Response: We have removed TSB3 from figure 5B.

Moreover, other genes involved in the conversion from IGP to indole should also be examined for expression changes. Reference: The biosynthesis of auxin: how many paths truly lead to IAA?. Plant Growth Regulation, 78(3): 275–285. Additionally, IAMH1/2 are currently considered the primary catalysts for IAM-to-IAA conversion. Have these genes been analyzed for expression changes? Reference: Two homologous INDOLE-3-ACETAMIDE (IAM) HYDRALASE genes are required for the auxin effects of IAM in Arabidopsis. Journal of Genetics and Genomics, 2020, 47(3).

Finally, while the authors validated selected CK-related genes, similar validation should be performed for auxin-related genes discussed in this section.

 Response: We have revised Figure 5 (see new panel C) to include qRT-PCR expression analyses of the following genes associated with IAA and GLS biosynthesis: CYP79B2, YUC3, YUC5, YUC6, YUC8, SUR1, SUR2, and AMI1. Although time and material limitations prevented us from testing every gene suggested by the reviewer, we believe that the additional data provide sufficient support, at least partially, for our RNA-seq analysis.

Minor Concerns:

1) The keywords are excessive and inadequately selected; please revise them.

Response: We believe that the selected keywords accurately reflect the players, phenotypes, and biological mechanisms addressed in this work.

2) The abbreviation "CK" for cytokinin should be used consistently throughout the manuscript instead of alternating between full and abbreviated forms.

Response: We have used the abbreviation for CK consistently in the revised text.

3) Gene symbols and mutant names should be italicized.

Response: corrected

4) In the Introduction, “…members of the YUCCA family and IAOx pathways” should be corrected to “…members of the IPyA pathway and IAOx pathway.”

Response: corrected

5) In the Results section, “indole-3-pyruvate (IPA)” is abbreviated as “IPyA” elsewhere. The authors should uniformly use “IPyA.”

Response: corrected

6) In the Results section, the statement “Conversely, yuc loss-of-function mutants exhibit auxin-deficiency symptoms…” requires a supporting citation, as only yuc8, a single mutation of YUC, has been clearly demonstrated to display notable defects under cytokinin conditions. Please cite: Functional roles of Arabidopsis CKRC2/YUCCA8 gene and the involvement of PIF4 in the regulation of auxin biosynthesis by cytokinin. Scientific Reports, 6, 36866.

Response: Our statement focuses on the redundant role of YUCCA genes in the auxin biosynthetic pathway (specifically, the rescue of high-order yuc mutant phenotypes by auxin application) and is supported by the appropriate references. We do not intend to imply a role for YUCCAs within the cytokinin pathway. Nevertheless, we have revised the sentence as follows:
“Conversely, yuc loss-of-function mutants exhibit auxin-deficiency symptoms, which can be partially restored through exogenous auxin supplementation or endogenous tissue-specific auxin production [44, 47–49].”

Round 2

Reviewer 3 Report

Comments and Suggestions for Authors

The authors have not fully addressed my concerns raised in the previous round of review, and their responses have unfortunately introduced new ambiguities:

1) Regarding Figure 3D, where the authors present up- and down-regulated genes associated with different hormonal pathways, the complete gene lists corresponding to these pathways should be provided as supplementary material.

2) In Figure 5A, TSB3 remains included despite previous comments. Furthermore, I previously noted that IAMH1/2 are considered the primary enzymes catalyzing the conversion of IAM to IAA. If these genes were detected in the RNA-seq analysis, any results concerning them should be reported. Additionally, the currently accepted pathway correctly describes the conversion from IAOx to IAN as previously depicted. The model should be revised according to the latest literature, specifically: Fenech, M.; Brumos, J.; Pěnčík, A.; Edwards, B.; Belcapo, S.; DeLacey, J.; Patel, A.; Kater, M.; Li, X.; Ljung, K.; et al. The CYP71A, NIT, AMI, and IAMH Gene Families Are Dispensable for Indole-3-Acetaldoxime-Mediated Auxin Biosynthesis in Arabidopsis. Plant Cell 2025, 37, koaf242, doi:10.1093/plcell/koaf242.

3) I note that in Figure 5C, the transcript levels of several genes presented by the authors, including YUC6 and YUC8, in the hat3 athb4 mutant are less than half of those in Col-0, yet are reported as not statistically significant. This is difficult to interpret and warrants careful re-examination by the authors.

4) The statement: "Arabidopsis thaliana possesses 11 YUC genes, and their overexpression has been shown to result in strong auxin-overproduction phenotypes [42–46]. Conversely, yuc loss-of-function mutants exhibit auxin-deficiency symptoms, which can be partially restored through exogenous auxin supplementation or endogenous tissue-specific auxin production [44,47–49]" should be supplemented with a citation to the study: "Functional roles of Arabidopsis CKRC2/YUCCA8 gene and the involvement of PIF4 in the regulation of auxin biosynthesis by cytokinin. Scientific Reports, 6, 36866." This reference identifies yucca8 as the single YUC mutant exhibiting a pronounced phenotype, which would strongly support the authors' argument.

Author Response

The authors have not fully addressed my concerns raised in the previous round of review, and their responses have unfortunately introduced new ambiguities:

1) Regarding Figure 3D, where the authors present up- and down-regulated genes associated with different hormonal pathways, the complete gene lists corresponding to these pathways should be provided as supplementary material.

Response: The lists of genes assigned to each GO category, including both up- and down-regulated genes classified under “Response to Hormone” and used to generate Figure 3D, are already provided in Supplementary Table 2, as cited in Results section 2.3.

2) In Figure 5A, TSB3 remains included despite previous comments. Furthermore, I previously noted that IAMH1/2 are considered the primary enzymes catalyzing the conversion of IAM to IAA. If these genes were detected in the RNA-seq analysis, any results concerning them should be reported. Additionally, the currently accepted pathway correctly describes the conversion from IAOx to IAN as previously depicted. The model should be revised according to the latest literature, specifically: Fenech, M.; Brumos, J.; Pěnčík, A.; Edwards, B.; Belcapo, S.; DeLacey, J.; Patel, A.; Kater, M.; Li, X.; Ljung, K.; et al. The CYP71A, NIT, AMI, and IAMH Gene Families Are Dispensable for Indole-3-Acetaldoxime-Mediated Auxin Biosynthesis in Arabidopsis. Plant Cell 2025, 37, koaf242, doi:10.1093/plcell/koaf242.

Response: We have removed TSB3 from the heatmap in Figure 5A.

Moreover, in our dataset IAMH1is downregulated (-2.1457logFC) and falls below the FDR cutoff 0.5, whereas IAMH2 is conversely upregulated (~0.435logFC). Because of their opposite behavior, we did not focus on that branch of the auxin biosynthesis pathway and therefore did not examine these genes further.

Finally, as written in the figure legend, in the auxin pathways shown in Fig. 5B we included only the genes identified from our RNA-seq analysis that were either upregulated (in red) or downregulated (in blue). This panel is intended to help the reader locate the corresponding enzymes within the different biosynthetic routes, and is not meant to serve as a comprehensive review of auxin biosynthetic pathways.

3) I note that in Figure 5C, the transcript levels of several genes presented by the authors, including YUC6 and YUC8, in the hat3 athb4 mutant are less than half of those in Col-0, yet are reported as not statistically significant. This is difficult to interpret and warrants careful re-examination by the authors.

Response: The statistical difference for YUC6 was P = 0.0535, while for YUC8 was P =0.0942. The qRT-PCR analyses for these two genes were based on only two biological replicates. As explained in our previous response, this limitation was due to restricted material availability and time constraints. Nevertheless, we believe that the expression trends of these two genes are consistent with the RNA-seq results.

We have added the following information to the caption of Figure 4 and 5: “N≥2”.

4) The statement: "Arabidopsis thaliana possesses 11 YUC genes, and their overexpression has been shown to result in strong auxin-overproduction phenotypes [42–46]. Conversely, yuc loss-of-function mutants exhibit auxin-deficiency symptoms, which can be partially restored through exogenous auxin supplementation or endogenous tissue-specific auxin production [44,47–49]" should be supplemented with a citation to the study: "Functional roles of Arabidopsis CKRC2/YUCCA8 gene and the involvement of PIF4 in the regulation of auxin biosynthesis by cytokinin. Scientific Reports, 6, 36866." This reference identifies yucca8 as the single YUC mutant exhibiting a pronounced phenotype, which would strongly support the authors' argument.

Response: We have added the aforementioned reference, as indicated by the reviewer (see reference 50).

Round 3

Reviewer 3 Report

Comments and Suggestions for Authors

The authors have answered all my concerns.